# Melt Conditioned Direct Chill (MC-DC) Casting and Extrusion of AA5754 Aluminium Alloy Formulated from Recycled Taint Tabor Scrap

**DOI:** 10.3390/ma13122711

**Published:** 2020-06-15

**Authors:** Kawther Al-Helal, Jayesh B. Patel, Geoff M. Scamans, Zhongyun Fan

**Affiliations:** 1Brunel Centre for Advanced Solidification Technology, Brunel University London, London UB8 3PH, UK; jayesh.patel@brunel.ac.uk (J.B.P.); zhongyun.fan@brunel.ac.uk (Z.F.); 2Innoval Technology Limited, Beaumont Cl, Banbury OX16 1TQ, UK; geoff.scamans@innovaltec.com

**Keywords:** recycling of aluminum alloys, direct chill casting, high shear melt conditioning, Taint Tabor scrap, extrusion, heat treatment, cold rolling

## Abstract

The melt conditioned direct chill (MC-DC) casting process has been used to produce billets and extruded planks of AA5754 alloy formulated from 100% recycled Taint Tabor scrap aluminum. The billets were homogenized and then extruded into flat planks. Optical metallography of the MC-DC cast billets showed equiaxed refined grains in comparison to conventional direct chill (DC) cast and direct chill grain refined (DC-GR) cast billets formulated from the same Taint Tabor scrap. Microstructural evaluation of the extruded planks showed extensive peripheral coarse grain (PCG) for the DC, DC-GR and MC-DC cast planks. The 2 mm and 1 mm MC-DC cast planks produced after cold rolling and heat treatment showed a fully recrystallized microstructure at 380 °C and 300 °C for 10 min respectively with an improvement in mechanical properties over DC-GR cast and similarly processed planks. The as-extruded MC-DC cast planks tensile tested in the transverse direction showed 34% elongation and 213 MPa ultimate tensile strength. These tensile results showed 5.8% higher elongation and 1.2% higher ultimate tensile strength compared with the DC-GR planks after applying high shear melt conditioning.

## 1. Introduction

Research activities on recycling to save natural resources have been well supported and funded for the past few years. The main target of this research has been the extraction of metallic fractions from waste and scrap sources to facilitate its reuse in industrial applications. Recycling of Al-alloys has an economic and environmental benefit as the processing of secondary aluminum from recycled resources consumes 2.8 kWh/kg while the production of the primary Al requires about 45 kWh/kg of energy [1].

AA5754 is an important automotive body sheet alloy used for structural applications in the automotive industry because of its good formability. Due to its deep drawing characteristics, it is used in structural panels as an alternative to a mild steel sheet. In addition, AA5754 alloy has excellent corrosion resistance and it is an alloy suitable for recycling [2,3]. In collaboration between Novelis UK and Jaguar Land Rover, a new automotive product RC5754 alloy has been developed. RC5754 alloy is designed to contain up to 75% recycled content and has been successfully integrated into the structural components of high volume production passenger vehicles. In this study, AA5754 alloy was formulated from 100 percent Taint Tabor (TT) recycled aluminum scrap supplied by Axion. The typical Taint Tabor (TT) in the UK consists of aluminum wrought scrap of two or more alloys that is free of castings, foil, wire, Venetian blinds or any other non-metallic items [4]. The TT supplied to Axion contains a large fraction of production scrap rather than post-consumer scrap. Prior to ingot casting, this TT is shredded and then processed through magnetic separation and then eddy current separation in addition to some hand sorting to remove obvious zinc castings.

Direct chill (DC) casting is the main industrial casting process for the production of semi-finished products such as billets or slabs for further processing by forging, rolling or extrusion. The main problems in DC casting are chemical segregation, formation of coarse grain with non-uniform microstructure, porosity and hot tearing [5]. Due to the limitations of using chemical grain refinement with Al-alloys, alternative physical grain refinement techniques have been investigated such as ultrasonic cavitation; electromagnetic stirring and the high shear melt conditioning (HSMC) technology [6,7]. The casting technology, known as melt conditioned direct chill (MC-DC) casting, was developed at the Brunel Centre for Advanced Solidification Technology (BCAST) for as-cast microstructural improvement and to eliminate common casting defects [8]. It has been tested for the production of large-scale billets by applying intensive melt shearing in the sump of the direct chill casting process [6]. It has demonstrated that the dispersed oxide particles generated by the high shear melt conditioning process act as nucleation sites for grains and iron-rich intermetallic compounds (IMC’s). HSMC technology can potentially overcome some of the limitations of conventional casting processes for recycled materials. The HSMC technology, which consists of a rotor-stator mixing device, has been considered as a multi-purpose device. It can be used as an alternative method for grain refinement, casting of metal-matrix composites, degassing of melt prior to casting and de-ironing for any industrial casting processes [6,7]. In this study, HSMC technology has been used for the casting of AA5754 alloy formulated from the recycled aluminum source. Recycling of aluminum alloys for industrial applications delivers cost savings and low CO_2_ emission.

The aim of this study was to investigate the impact of HSMC on the microstructure and mechanical properties of the AA5754 alloy formulated from 100% Taint Tabor (TT) scrap aluminum processed into a flat bar after DC, DC-GR and MC-DC casting followed by subsequent thermomechanical treatment of extrusion and cold rolling.

## 2. Experimental Procedures

A one tone batch of processed Taint Tabor scrap from Axion was melted and cast into standard 9 kg ingots by Norton Aluminum and was supplied to BCAST for this research activity. The ingots were melted and the melt composition was adjusted by additions of Mn and Mg to match the standard chemical composition of the AA5754 alloy. The compositions of as received TT ingots, the formulated TT-5754 and standard AA5754 are given in Table 1. By using a Thermo-Analyze machine IDECO TA 748CT, the liquidus temperature of the formulated TT-5754 was measured to be ~912 K (639 °C).

For casting of TT-5754 alloy, a conventional DC caster was used. It consisted of a big tilting furnace, a launder and an open-hot-top assembly. The casting speed was 178 mm/min with a water flow rate equal to 180 L/min. Billets of ~2 m length and 152 mm diameter were cast successfully. Al-5Ti-1B grain refiner was added to the melt in the launder before pouring into the DC casting system for the direct chill grain refined casting, the melt temperature in the launder was 943 K (670 °C). In melt conditioned direct chill casting, the rotor-stator shearing device was preheated to ~700 °C and mounted on top of the open hot-top of the system. The shearing device was lowered in the melt and switched on to start at ~2000 rpm as soon as a steady-state casting speed of 178 mm/min was achieved. An electric controller was used to adjust the position of the shearing device in the sump. The DC, DC-GR and MC-DC casting and the subsequent extrusion and thermomechanical processes were all carried out at the Advanced Metal Casting Centre (AMCC) at Brunel University London. The schematic of the MC-DC casting process, the billets produced and the extruded flat bars are shown in Figure 1.

The TT-5754 alloy billets were homogenized at 530 °C for 4 h followed by air cooling and extruded at a temperature of 530 °C and 3 m/min speed into a flat bar of 118 mm width by 4.8 mm thickness. The extrusion line at AMCC had a 16 MN long stroke direct extrusion press. The billet dimensions required for extrusion are 152 mm diameter and 450 mm length. The billet heater and quench had taper control and post extrusion handling consist of press lead-out, quench, 14 m run-out table, stretcher, finishing saw and cut to length table.

### 2.1. Microstructure Characterization

Metallographic characterizations were carried out on samples from billets at different positions and from extrusion planks in the transverse and longitudinal orientations. These samples were mounted in Bakelite for grounding and polished down to a 0.04 µm silica in water suspension. For grain size and grain morphology characterization, the polished samples were anodized at 20 V for 55 s in Barker’s reagent. A Zeiss Axio-Vision optical microscope was used for analyzing the DC, DC-GR and MC-DC cast samples for all casting conditions and orientations. The average grain size on polarized images was measured by using the linear intercept method. A JCM-6000PLUS BENCHTOP SEM was used for the scanning electron microscopy (SEM) examination of the samples with an accelerating voltage of 10 kV.

### 2.2. Thermomechanical Treatment and Tensile Testing

For the thermomechanical processing, samples from the as-cast planks were taken in the middle of extruded lengths to ensure that front end and back-end defects were eliminated. The 4.8 mm gauge planks were homogenized at 520 °C for 8 h followed by air cooling to room temperature. Deformation by cold rolling was used to reduce the thickness of the homogenized strips to 2 mm and 1.0 mm gauge. Cold work was relieved by intermediate annealing heat treatments. The minimum temperature for full recrystallization for the two gauges was investigated by metallographic characterization.

Flat tensile specimens were machined from the as-extruded bar and from the cold-rolled and annealed strips from the DC-GR and MC-DC cast planks. Tensile specimen dimensions were in accordance with British Standard EN ISO 6892-1 as shown in Figure 2.

The tensile properties were measured on samples cut in 0°, 90° and 45° orientation for as-extruded planks and from the 0° and 90° orientations for the thermomechanical treated samples. At least five samples for each casting and heat treatment conditions were used for the tensile tests. An Instron 5500 universal 50 kN electromechanical testing system was used for tensile testing with the dual-rate method at room temperature. The rate 1 was 0.01 strain/min to 1% strain and the rate 2 was 0.4 strain/mm to failure.

## 3. Results and Discussion

### 3.1. Direct Chill Casting

Figure 3 shows the optical micrographs TT-5754 alloy DC, DC-GR and MC-DC cast billets. The microstructure of DC-GR and MC-DC cast billets showed equiaxed refined grains with uniform grain size distribution in comparison with the DC cast billet. The average grain sizes for DC, DC-GR and MC-DC at the half radius position were 350, 220 and 190 µm respectively. Casting of AA5xxx aluminum alloys and applying high shear melt conditioning results in significant grain refinement due to the presence of oxides of magnesium in this alloy system. Films and discrete oxide particles are the types of the oxides formed as the melt surface is entrained into the casting [8]. Due to the physical action of the intensive melt shearing, these oxides break and disperse throughout the liquid uniformly [9]. The dispersed oxide particles can enhance the nucleation process and nucleate α-Al phase [10]. This enhanced nucleation process leads to the formation of the fine equiaxed grain structure across the billet with the elimination of the segregation of solutes to the surface of the billet.

Figure 4 shows the phase diagram of the formulated TT-5754-xSi alloy at equilibrium, calculated using PANDAT software. In casting TT-5754 with 0.52 wt.% Si, the intermetallic phases formed are α-Al_15_(Fe, Mn)_3_Si_2_, Mg_2_Si and Al_6_FeMn. In Al-Si-Mg alloys with Mn, the structure of Fe-rich intermetallic α-Al_15_(Fe, Mn)_3_Si_2_ is body-centered cubic, which appears as hexagonal, star-shaped, or dendritic crystals at different Mn/Fe ratios. Morphological control of this intermetallic from plate-like to compact shapes or Chinese script can reduce its detrimental effect on final mechanical properties [11]. The formation of Fe-rich phases can block the interdendritic flow regions and result in increased porosity and formation of Fe-rich intermetallic is strongly affected by solidification conditions [12].

The metallographic analysis using scanning electron microscopy (SEM) with EDS analysis in Figure 5 shows the presence of α-Al solid solution matrix with Chinese script like morphology Mg_2_Si precipitates as the primary dendrites that were located on grain boundaries in the interdendritic regions. Additionally, it was found that the microstructure also contained precipitates of Al_15_(Fe, Mn)_3_Si_2_ with an irregular shape with no evidence of the Al_6_FeMn intermetallic compound. Figure 5 shows that the size and morphology of these intermetallics changed with the casting conditions. Applying intensive melt shearing refined these intermetallics and changed the morphology of Al_15_(Fe, Mn)_3_Si_2_ from Chinese script to compact shapes that solidified within the Mg_2_Si precipitate.

### 3.2. Extruded Planks

The optical micrographs of the TT-5754 extruded planks are shown in Figure 6. All casting variants showed extensive peripheral coarse grain (PCG). Peripheral coarse grain is a layer of recrystallized coarse grains that form in the outer band of an extruded section and is a well-known phenomenon [13]. PCG is generally observed in aluminum alloys such as AA5xxx series alloys and is a surface imperfection that results in low machinability, mechanical properties and poor surface appearance [14]. The degree of PCG depends on the alloy composition, billet microstructure, presence of Mn, Zr and Cr (grain growth inhibitors) in addition to extrusion conditions such as billets temperature and extrusion speed [15,16].

### 3.3. Tensile Properties and Thermomechanical Treatments

#### 3.3.1. As-Extruded Planks

The tensile properties of as-extruded DC-GR and MC-DC planks obtained from homogenized billets of the formulated TT-5754 alloy are shown in Figure 7 and Table 2. The DC-GR and MC-DC results show similar elongation and ultimate tensile strength in all the orientations tested. The yield strength ratio was also calculated for all casting conditions and orientations. It is the ratio of the yield strength and ultimate tensile strength of the material. The higher ratio is an indication for the better anti-deformation ability and the lower ratio for the better plasticity [17]. The results in Table 2 show that the yield strength ratio for MC-DC was similar to that for DC-GR.

#### 3.3.2. Cold Rolling with Heat Treatment

##### Cold Rolling to a 2 mm Gauge with Heat Treatment

Figure 8 shows the optical micrographs of the cold-rolled TT-5754 strip. The microstructure of the cold-rolled extruded planks showed an un-recrystallized fibrous structure core with no peripheral coarse grain (PCG) for all casting variants. The minimum annealing temperature for full recrystallization of the 2 mm gauge strip was between 350 °C and 380 °C as shown in Figure 9 and Figure 10. There was full recrystallization for all casting conditions at 380 °C for 10 min as shown in Figure 10.

Figure 11 shows that annealing at a higher temperature of 520 °C for 3 h, the 2 mm gauge of formulated TT-5754 alloy has undergone complete recrystallization, forming fine recrystallized grains with the fibrous microstructure completely disappeared. The tensile test was carried out for samples annealed at 380 °C for 10 min and the results are listed in Table 3 and shown in Figure 12.

The results of tensile tests in the longitudinal orientation show very similar tensile test results for all the casting conditions after cold rolling and heat treatment once the as-cast microstructure had been fully transformed. While, in the transverse orientation, there was a significant improvement in the tensile results after applying melt conditioning in comparison with the DC-GR condition.

##### Cold Rolling to a 1 mm Gauge with Heat Treatment

Figure 13a–c shows the optical micrographs of cold rolled TT-5754 alloy strip from 4.8 mm to 1 mm gauge for DC, DC-GR and MC-DC cast samples. The microstructure of the cold rolled strip showed an un-recrystallized fibrous structure core with no peripheral coarse grain (PCG) for all casting variance. Annealing heat treatment conditions for all samples were tested for a wide range of temperatures (200–520 °C) and time (10 min–3 h). The microstructures for all samples were checked in order to find out the minimum temperature and time for full recrystallization. In the range of 200 °C to 520 °C for 10 min heat treatment, the optical micrographs show that the minimum temperature for annealing 1 mm gauge strip was between 290 °C to 300 °C as shown in Figure 13d–i and Figure 14. It is very clear that in the transverse and longitudinal directions there was full recrystallization for all casting conditions at 300 °C for 10 min as shown in Figure 14.

Optical micrographs in Figure 15 and Figure 16 show that after annealing at higher temperatures like 400 °C for 10 min and at 520 °C for 3 h, the 1 mm gauge of formulated TT-5754 alloy has finished the recrystallization and forming very fine recrystallized grains. The tensile test was carried for samples annealed at 300 °C and 400 °C for 10 min in order to investigate the effect of annealing temperature on the tensile properties.

Figure 17 and Table 4 show the tensile properties of a 1 mm gauge strip after cold rolling and annealing at 300 °C and 400 °C for 10 min. The results show little increase of elongation and ultimate tensile strength with a decrease in yield strength for MC-DC in 90° orientation in comparison to DC-GR after annealing at 300 °C for 10 min. While for annealing at 400 °C, the elongation increases to 21.8% for the MC-DC samples compared with 18.2% for DC-GR and the ultimate tensile strength decreased to 229 MPa for MC-DC samples compared with 255 MPa for DC-GR. This means that MC-DC showed 20% higher elongation with 10% lower ultimate tensile strength compared with DC-GR and the yield strength decreased by 2.2%. The yield strength ratio in Table 3 and Table 4 show that better plasticity can be achieved with the increase of annealing temperature of 1 mm gauge of formulated TT-5754 alloy for all casting conditions. Once again, the results of all the mechanical tests are very similar for all the tested variants once the as-cast microstructure has been transformed by the recrystallization treatment.

With the increase of annealing temperature, the tensile strength and yield strength of the 1 mm gauge of TT-5754 alloy decreased, and elongation increased. This is mainly due to large plastic deformation that the 1 mm cold rolled planks have experienced and a large amount of deformation energy stored. The strength reduction with plasticity increase after annealing is the result of recovery and recrystallization which offset the deformation during the hardening process [17]. In order to achieve mechanical properties of developed RC5754 listed in Table 5 [18], the annealing temperature of 1 mm extruded TT-5754 will be in the range of 300–400 °C for 10 min.

## 4. Conclusions

The microstructure of DC-GR and MC-DC cast billets showed equiaxed refined grains with uniform grain size distribution in comparison to DC cast without the addition of a grain refiner. The average grain size for DC, DC-GR and MC-DC cast at the half radius position were 350, 220 and 190 µm respectively.The metallographic analysis of billet samples using scanning microscopy SEM with EDS analysis revealed the presence of α-Al solid solution matrix with Chinese-script-like morphology Mg_2_Si precipitates as the primary dendrites in addition to precipitates of Al_15_(Fe, Mn)_3_Si_2_ having an irregular shape. Applying intensive melt conditioning refined these intermetallics and changed the morphology of Al_15_(Fe, Mn)_3_Si_2_ from Chinese script to compact shapes that solidified within the Mg_2_Si precipitates.The as extruded planks of TT-5754 had extensive peripheral coarse grain (PCG) for all the DC, DC-GR and MC-DC casting conditions.The tensile properties of the as extruded DC-GR and MC-DC cast extruded planks were similar.The microstructure of 1 and 2 mm cold rolled planks showed an un-recrystallized fibrous structure core with no peripheral coarse grain (PCG) for all casting variance. The recrystallization temperature of the 1 mm gauge strip was 300 °C, while for a 2 mm gauge it was 380 °C.The tensile properties results of a 2 mm gauge for all the strip variants tested were similar.Annealing of a 1 mm gauge at 400 °C, the elongation increased to 21.84% for the MC-DC cast samples compared with 18.20% for DC-GR cast and the ultimate tensile strength decreased to 228.92 MPa for MC-DC cast samples compared with 255.04 MPs for DC-GR casting. This means that MC-DC casting showed 20% higher elongation with 10% lower ultimate tensile strength compared with DC-GR casting and the yield strength decreased by 2.2%.With the increasing annealing temperature, the tensile strength and yield strength of the 1 mm gauge strip decreased, and elongation increased.

## Figures and Tables

**Figure 1 materials-13-02711-f001:**
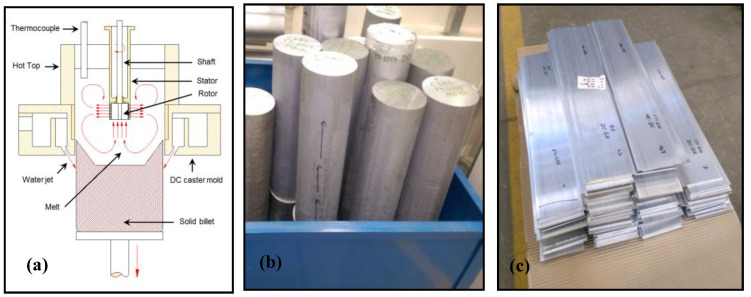
(**a**) Schematic diagram of the melt conditioned direct chill (MC-DC) casting technology, (**b**) billets of TT-5754 alloy, and (**c**) extruded planks produced from direct chill (DC), direct chill grain refined (DC-GR) and MC-DC cast billets of TT-5754 alloy.

**Figure 2 materials-13-02711-f002:**
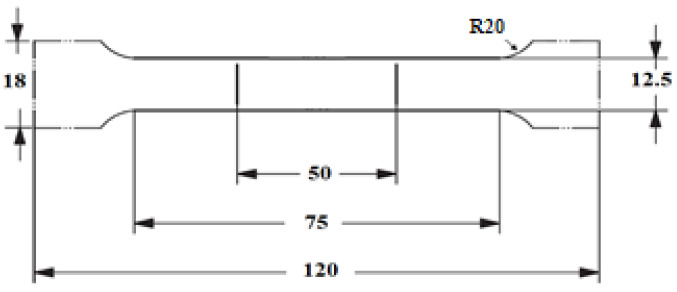
Tensile test specimen dimensions according to BS EN ISO 6892-1:2016 (all dimensions are in mm).

**Figure 3 materials-13-02711-f003:**
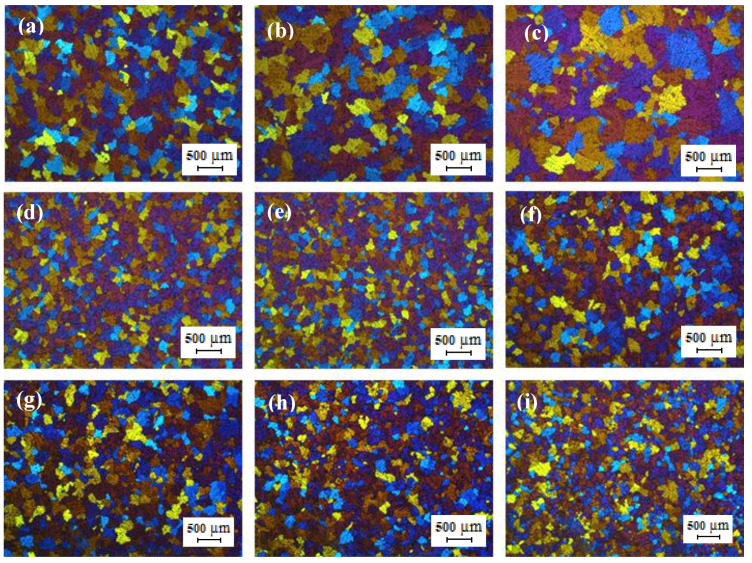
Optical micrographs of direct chill cast billets of formulated TT-5754 alloy at the edge, half radius and center (left to right): (**a**–**c**) DC; (**d**–**f**) DC-GR and (**g**–**i**) MC-DC.

**Figure 4 materials-13-02711-f004:**
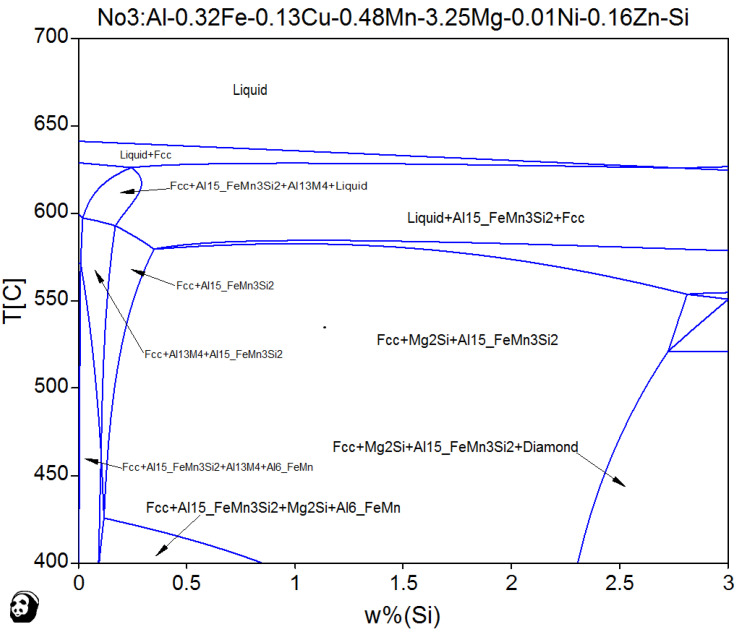
Equilibrium phase diagram of formulated TT-5754-xSi.

**Figure 5 materials-13-02711-f005:**
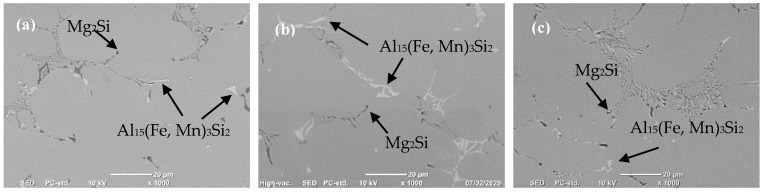
SEM micrographs of direct chill cast billets of TT-5754 alloy (**a**) DC; (**b**) DC-GR and (**c**) MC-DC.

**Figure 6 materials-13-02711-f006:**
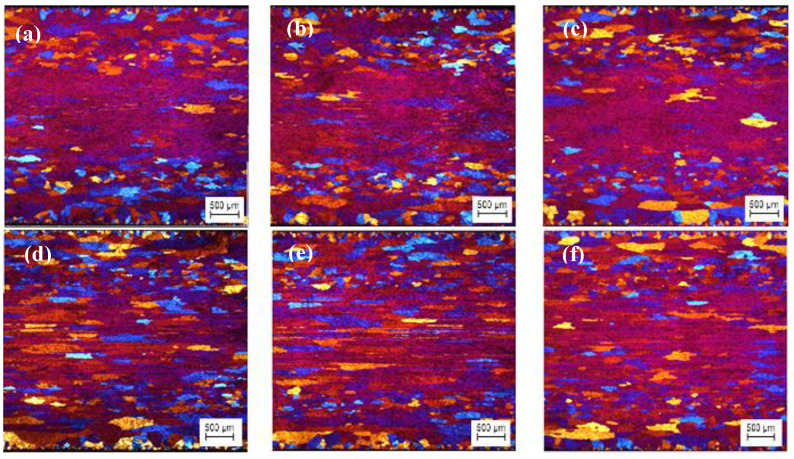
Optical micrographs of planks extruded from homogenized TT-5754 billets; (**a**,**d**) DC; (**b**,**e**) DC-GR and (**c**,**f**) MC-DC; (**a**–**c**) transverse direction and (**d**–**f**) longitudinal direction.

**Figure 7 materials-13-02711-f007:**
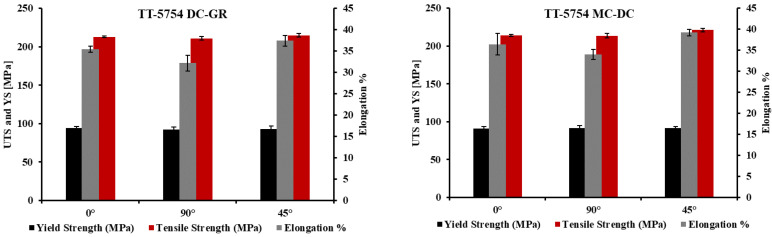
Plots of tensile properties of the formulated TT-5754 planks extruded from homogenized DC-GR and MC-DC cast billets.

**Figure 8 materials-13-02711-f008:**
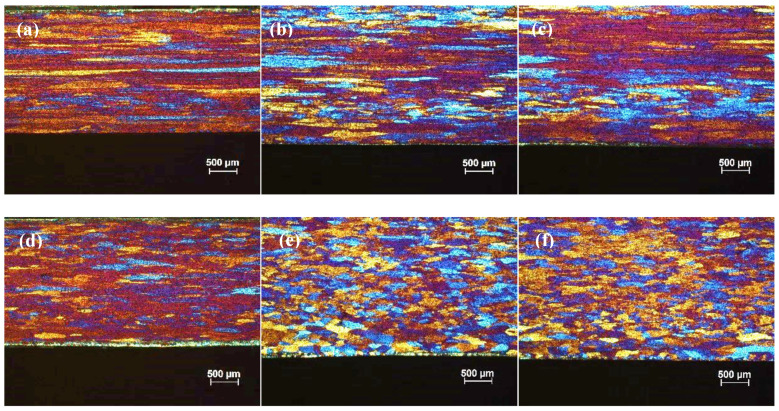
Optical micrographs of 2 mm cold rolled TT-5754 alloy planks; (**a**,**d**) DC; (**b**,**e**) DC-GR and (**c**,**f**) MC-DC; (**a**–**c**) longitudinal direction and (**d**–**f**) transverse direction.

**Figure 9 materials-13-02711-f009:**
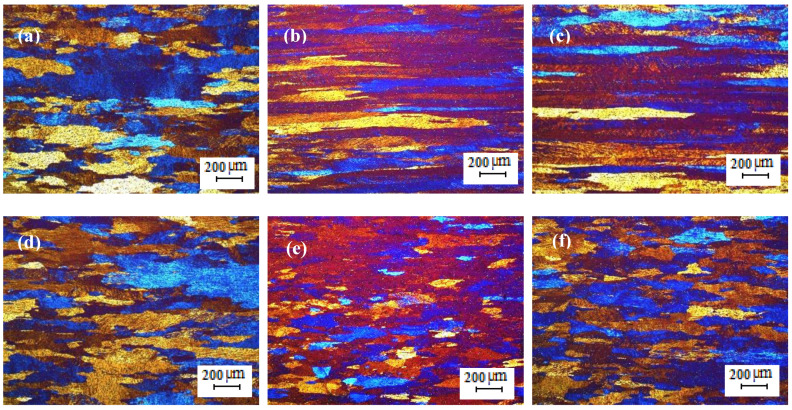
Optical micrographs of 2 mm cold rolled TT-5754 alloy planks annealed at 350 °C for 10 min.; (**a**,**d**) DC; (**b**,**e**) DC-GR and (**c**,**f**) MC-DC; (**a**–**c**) longitudinal direction and (**d**–**f**) transverse direction.

**Figure 10 materials-13-02711-f010:**
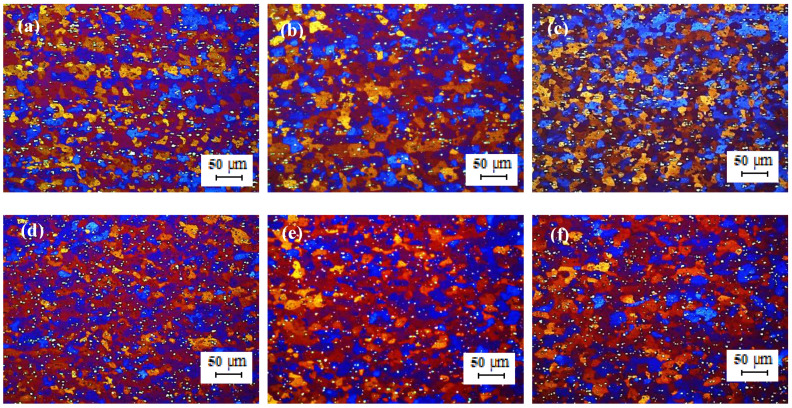
Optical micrographs of 2 mm cold rolled TT-5754 alloy planks annealed at 380 °C for 10 min; (**a**,**d**) DC; (**b**,**e**) DC-GR and (**c**,**f**) MC-DC; (**a**–**c**) longitudinal direction and (**d**–**f**) transverse direction.

**Figure 11 materials-13-02711-f011:**
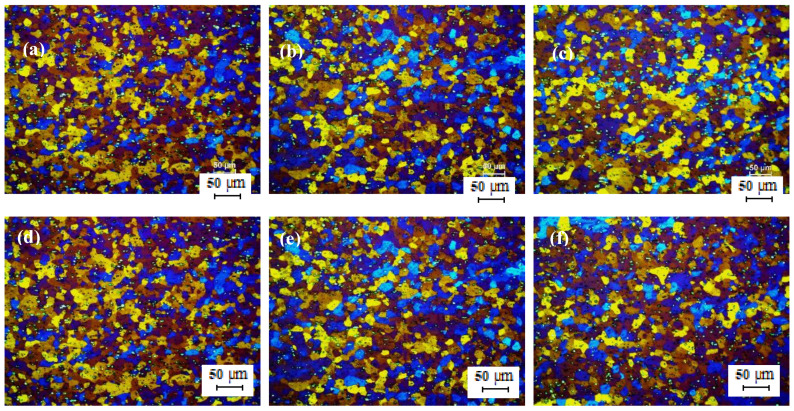
Optical micrographs of 2 mm cold rolled TT-5754 alloy planks annealed at 520 °C for 3 h; (**a**,**d**) DC; (**b**,**e**) DC-GR and (**c**,**f**) MC-DC; (**a**–**c**) longitudinal direction and (**d**–**f**) transverse direction.

**Figure 12 materials-13-02711-f012:**
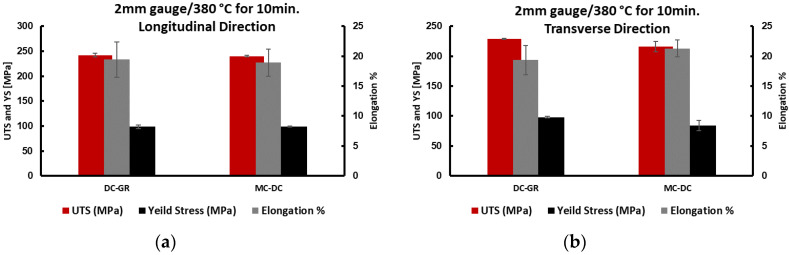
Tensile properties of 2 mm planks of formulated TT-5754 alloy annealed at 380 °C for 10 min; (**a**) longitudinal and (**b**) transverse sections.

**Figure 13 materials-13-02711-f013:**
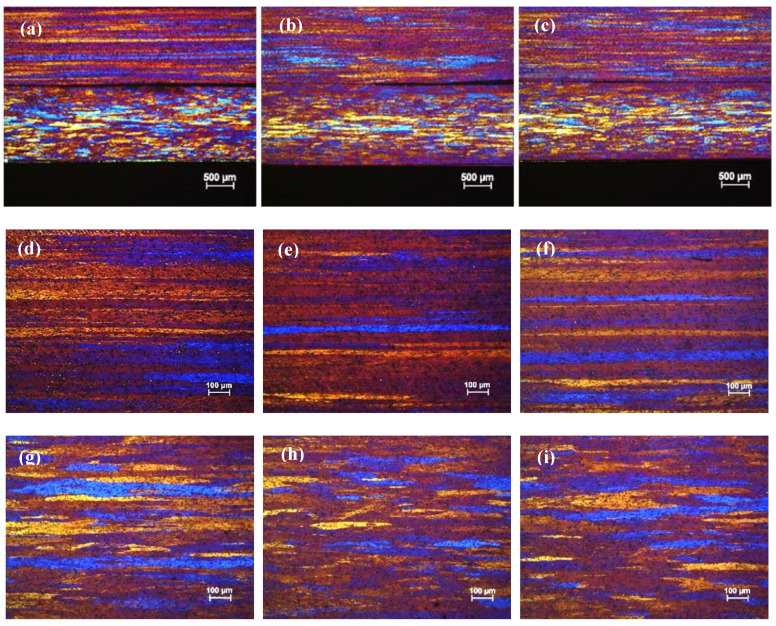
Optical micrographs of 1 mm cold rolled TT-5754 alloy planks; (**a**–**c**) as cold rolled in longitudinal (**top**) and transverse (**bottom**) directions for DC, DC-GR and MC-DC respectively; (**d**–**i**) annealed at 290 °C for 10 min, where (**d**,**g**) DC; (**e**,**h**) DC-GR and (**f**,**i**) MC-DC; (**d**–**f**) longitudinal direction and (**g**–**i**) transverse direction.

**Figure 14 materials-13-02711-f014:**
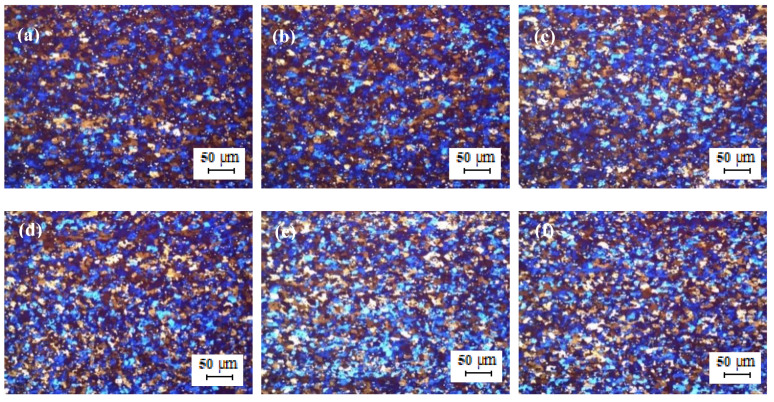
Optical micrographs of 1 mm cold rolled TT-5754 alloy planks annealed at 300 °C for 10 min; (**a**,**d**) DC; (**b**,**e**) DC-GR and (**c**,**f**) MC-DC; (**a**–**c**) longitudinal direction and (**d**–**f**) transverse direction.

**Figure 15 materials-13-02711-f015:**
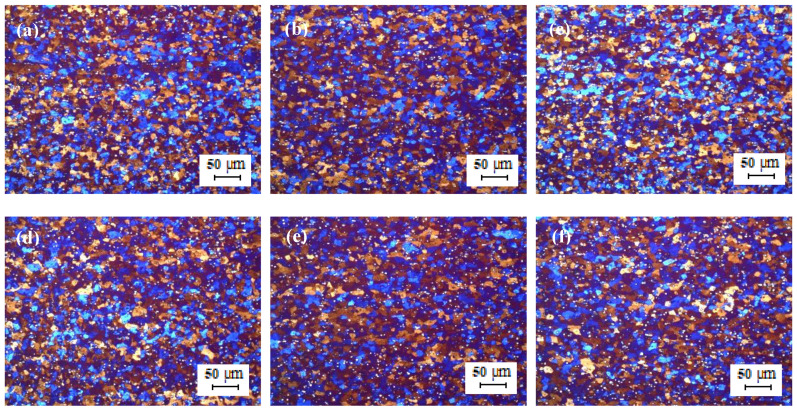
Optical micrographs of 1 mm cold rolled TT-5754 alloy planks annealed at 400 °C for 10 min; (**a**,**d**) DC; (**b**,**e**) DC-GR and (**c**,**f**) MC-DC; (**a**–**c**) longitudinal direction and (**d**–**f**) transverse direction.

**Figure 16 materials-13-02711-f016:**
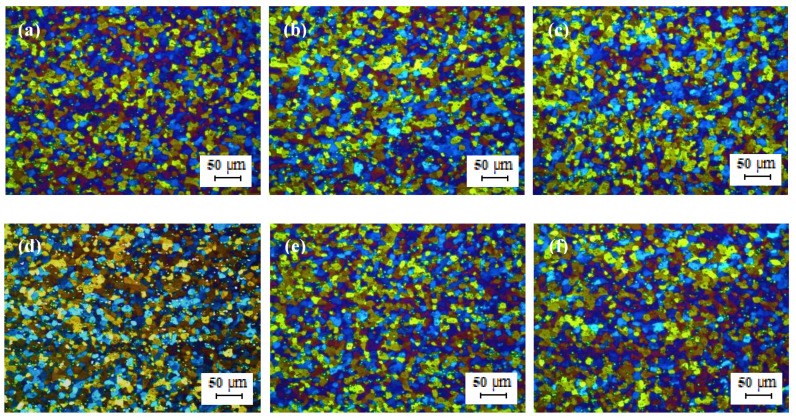
Optical micrographs of 1 mm cold rolled TT-5754 alloy planks annealed at 520 °C for 3 h; (**a**,**d**) DC; (**b**,**e**) DC-GR and (**c**,**f**) MC-DC; (**a**–**c**) longitudinal direction and (**d**–**f**) transverse direction.

**Figure 17 materials-13-02711-f017:**
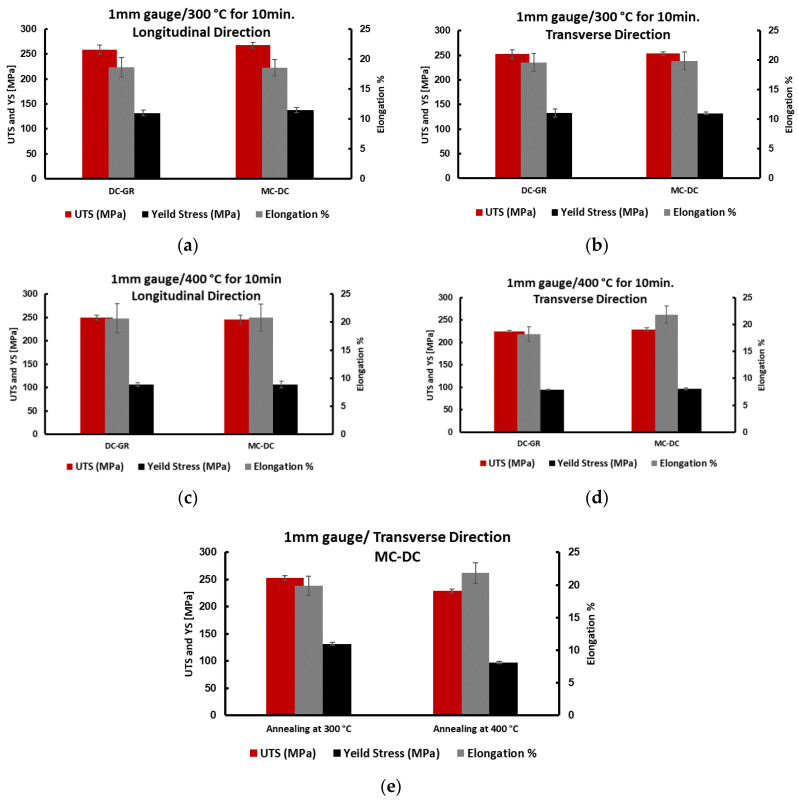
The mechanical properties of 1 mm planks of formulated TT-5754 alloy in longitudinal and transverse orientation; (**a**,**b**) annealed at 300 °C for 10 min; (**c**,**d**) annealed at 400 °C for 10 min and (**e**) Comparison at 300 °C and 400 °C for the MC-DC transverse section.

**Table 1 materials-13-02711-t001:** Chemical compositions of as received Taint Tabor (TT) ingots formulated TT-5754 and standard AA5754 alloy.

Material/wt.%	Si	Fe	Cu	Mn	Mg	Cr	Ni	Zn	Ti	Pb	Sn	Zr	Al
As received TT ingots	0.69	0.46	0.21	0.23	0.56	0.03	0.01	0.25	0.02	0.02	0.01	−	Bal.
Formulated TT-5754	0.52	0.32	0.13	0.48	3.25	0.02	0.01	0.16	0.02	0.004	0.001	0.001	Bal.
Standard AA5754	0.35	0.35	0.15	0.55	3.23	0.15	0.05	0.25	0.15	0.05	0.05	−	Bal.

**Table 2 materials-13-02711-t002:** Tensile properties of TT-5754 as-extruded planks at 0°, 90°, 45° orientation.

Casting Condition	DC-GR			MC-DC		
Sample direction	0°	90°	45°	0°	90°	45°
Yield Strength (MPa)	94	92	93	91	92	92
Tensile Strength (MPa)	213	211	215	214	213	221
Elongation%	35.4	32.1	37.4	36.4	34.0	39.2
Yield Strength Ratio	0.44	0.44	0.43	0.43	0.43	0.42

**Table 3 materials-13-02711-t003:** Tensile properties of 2 mm extruded planks of TT-5754 alloy annealed at 380 °C for 10 min.

Casting Condition	DC-GR		MC-DC	
sample direction	0°	90°	0°	90°
yield strength (MPa)	98	98	99	84
tensile strength (MPa)	242	229	240	216
elongation%	19.4	19.3	18.9	21.3
yield strength ratio	0.41	0.43	0.41	0.39

**Table 4 materials-13-02711-t004:** Tensile properties of 1 mm extruded planks of TT-5754 alloy annealed at 300 °C and 400 °C for 10 min.

Annealing Condition	Annealing at 300 °C for 10 min				Annealing at 400 °C for 10 min			
Casting Condition	DC-GR		MC-DC		DC-GR		MC-DC	
Sample Direction	0°	90°	0°	90°	0°	90°	0°	90°
Yield Strength (MPa)	132	133	138	132	106	95	106	97
Tensile Strength (MPa)	259	252	268	253	249	255	246	229
Elongation%	18.6	19.6	18.6	19.9	20.7	18.2	20.8	21.8
Yield Strength Ratio	0.51	0.53	0.52	0.52	0.43	0.37	0.43	0.42

**Table 5 materials-13-02711-t005:** Tensile properties of RC5754 Novelis [18].

Temper	Yield Strength (MPa)	Tensile Strength (MPa)	Elongation%
O/H111	105–145	220–260	≥20%

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
