# Peer review of "Melt Conditioned Direct Chill (MC-DC) Casting and Extrusion of AA5754 Aluminium Alloy Formulated from Recycled Taint Tabor Scrap"

_materials, 2020, doi:10.3390/ma13122711_

Round 1
Reviewer 1 Report
In this work, various direct chill casting methods have been applied to produce AA5754 billets and extruded planks using recycled scrap of AA5754. The grain sizes have been characterised and compared. The mechanical properties have been tested.
The work looks quite systematic and contains important knowledge and database for the field.
The questions the reviewer may raise here are:
(a) the authors claim their SEM work with EDS analysis. however, there is no SEM-EDS result included in the current version. Such results should be provided for the evidence of intermetallics.
(b) the title of this paper should be reconsidered to fit the content of the paper.
Author Response
Dear Reviewer,
I am very grateful to you and for the valuable suggestions provided in reviewing our manuscript.
Here are our responses to your comments:
(a) The authors claim their SEM work with EDS analysis. however, there is no SEM-EDS result included in the current version. Such results should be provided for the evidence of intermetallics.
Ans.: Unfortunately, due to lockdown all EDS data are on memory slick in my office. I may publish all EDS results including morphology and distribution of intermetallics for all casting conditions after lockdown.
(b) The title of this paper should be reconsidered to fit the content of the paper.
Ans.: We think that the manuscript title is exactly match the content. The alternative title that may fit the content of the paper could be the following:
Extrusion and thermomechanical treatment of Melt conditioned direct chill casting billets of AA5754 Aluminium alloy formulated from recycled Taint Tabor scrap
But it is too long.
Kind regards
Reviewer 2 Report
Dear Authors,
Congratulations on your sound research.
My only remarks are regarding fig. 7: the overlapping of the bars hide somewhat the error bars for the yield strength, and the stress-strain diagrams should be presented separately or even removed (low resolution of the image).
Also on the y axis the label is misleading: UTS (Yield Strength) MPa. Just use Strength [MPa] or UTS and YS [MPa], same observation for fig. 12 and 17.
Also, on line 273 "Annealing heat treatment for all samples were tested for a wide range of temperature and time." this sentence does not make sense to me.
My best regards.
Author Response
Dear Reviewer,
I am very grateful to you and for the valuable suggestions provided in reviewing our manuscript.
Here are our responses to your comments:
(a)-My only remarks are regarding fig. 7: the overlapping of the bars hide somewhat the error bars for the yield strength, and the stress-strain diagrams should be presented separately or even removed (low resolution of the image).
Ans.: The bars separated and the error bars are clear now for Fig. 7. See line 208.
(b)-Also on the y axis the label is misleading: UTS (Yield Strength) MPa. Just use Strength [MPa] or UTS and YS [MPa], same observation for fig. 12 and 17.
Ans.: Revised as requested for Fig. 7, 12 and 17. See Lines 208, 265 and 323.
(c)-Also, on line 273 "Annealing heat treatment for all samples were tested for a wide range of temperature and time." this sentence does not make sense to me.
Ans.: Annealing heat treatment conditions for all samples were tested for a wide range of temperature (200°C-520°C) and time (10min-3hrs). The manuscript revised as requested. See line 273.
Kind regards
Reviewer 3 Report
A very interesting scientific paper on an important practical issue.
The paper describes the reused of a scrap aluminium, which is a currently important topic. The authors give important information to have much energy can be saved by using the scrap.
Additionally, it was proved that mechanical properties and chemical composition can be similar to the production of the primary Al. It is important to notice that the authors describe in detail the technology of production. The experimental procedure is correctly presented, except for the tensile test, where he authors should define the traverse travel speed in mm/s.
Specific comments are the following:
In Fig. 2 it should be a dimension of radius R20, not 20.
Author's should give the date of viewing site for citation 18.
Author Response
Dear Reviewer,
I am very grateful to you and for the valuable suggestions provided in reviewing our manuscript.
Here are our responses to your comments:
Q1-Additionally, it was proved that mechanical properties and chemical composition can be similar to the production of the primary Al. It is important to notice that the authors describe in detail the technology of production. The experimental procedure is correctly presented, except for the tensile test, where he authors should define the traverse travel speed in mm/s.
Ans.: Sorry, I am little bit confused. I think travel speed is a welding parameter. I did tensile test for machined specimens. If you mean tensile method, it was dual rate. The rate 1 was 0.015 strain/min to 1% strain and the rate 2 was 0.400 strain/mm to failure. The manuscript revised as requested. See Line 130.
Q2-Specific comments are the following: In Fig. 2 it should be a dimension of radius R20, not 20.
Ans.: The manuscript revised as requested. See line 132.
Q3-Author's should give the date of viewing site for citation 18.
Ans.: The manuscript revised as requested. See line 428.
Kind Regareds